# Hyperhydricity in Plant Tissue Culture

**DOI:** 10.3390/plants11233313

**Published:** 2022-11-30

**Authors:** Oksana B. Polivanova, Vladislav A. Bedarev

**Affiliations:** Department of Biotechnology, Russian State Agrarian University Moscow Timiryazev Agricultural Academy, 49 Timiryazevskaya Street, Moscow 127550, Russia

**Keywords:** hyperhydricity, oxidative damage, *in vitro* culture, cytokinins, ethylene, clonal micropropogation, aeration, nitrogen ions

## Abstract

Hyperhydricity is the most common physiological disorder in *in vitro* plant cultivation. It is characterized by certain anatomical, morphological, physiological, and metabolic disturbances. Hyperhydricity significantly complicates the use of cell and tissue culture in research, reduces the efficiency of clonal micropropagation and the quality of seedlings, prevents the adaptation of plants in vivo, and can lead to significant losses of plant material. This review considers the main symptoms and causes of hyperhydricity, such as oxidative stress, impaired nitrogen metabolism, and the imbalance of endogenous hormones. The main factors influencing the level of hyperhydricity of plants *in vitro* are the mineral and hormonal composition of a medium and cultivation conditions, in particular the aeration of cultivation vessels. Based on these factors, various approaches are proposed to eliminate hyperhydricity, such as varying the mineral and hormonal composition of the medium, the use of exogenous additives, aeration systems, and specific lighting. However, not all methods used are universal in eliminating the symptoms of hyperhydricity. Therefore, the study of hyperhydricity requires a comprehensive approach, and measures aimed at its elimination should be complex and species-specific.

## 1. Introduction

During *in vitro* cultivation, plants are exposed to special conditions such as high humidity, insufficient lighting, excess sugars, and nitrates, which can lead to anatomical, morphological, and physiological disorders [1]. Hyperhydricity (vitrification, glassiness) is the most common physiological disorder associated with *in vitro* cultivation of plants [2]. Hyperhydric (glassy) shoots are characterized by thickened stems, short internodes, and translucent, fragile, elongated, and twisted leaves [2,3]. Symptoms of hyperhydricity also include chlorophyll deficiency, accumulation of large starch granules in plastids, cell hydration caused by the presence of excess fluid in the intercellular spaces, a decrease in cell adhesion and the formation of large intercellular spaces in the mesophil, hypolignification, a decrease in the formation of the epicuticular layer on leaf surfaces, changes in enzyme activity, and changes in protein synthesis due to disruption of normal metabolic processes [4,5,6,7,8,9,10]. Compared to normal plants, hyperhydric plants also exhibited damaged membranes and cell structure, reduction of cell wall thickness, decreased number of mitochondria, more intercellular spaces, higher cell vacuolation, and collapse of vascular tissues [11]. Disturbances of tissue structure and biochemical changes in hyperhydric plants are reflected in two main physiological processes: gas exchange and photosynthesis. Disorders associated with hyperhydricity appear mainly on leaves but also on stems and roots. Hyperhydricity has been studied in most detail in carnation microclones [4,7,12,13]. Carnation was chosen as a model object for studying hyperhydricity since it effectively micropropagates *in vitro* and hyperhydricity can be easily induced in a given culture [13]. To date, hyperhydricity symptoms have been observed in about 200 plants cultivated *in vitro* [1,2,3,4,14,15]. Some species for which hyperhydricity has been noted during *in vitro* cultivation and effective approaches for eliminating its symptoms are presented in Table 1.

In addition, hyperhydricity complicates agrobacterium-mediated transformation and *in vitro* regenerations, the conservation of rare and endangered plant species using *in vitro* methods, and generally limits the use of plant cell and tissue culture for basic research and practical use. To date, there is no consensus regarding the causes of hyperhydricity, and measures to eliminate it are species-specific. The purpose of this review is to characterize hyperhydricity through the anatomical, morphological, and biochemical features underlying this disorder; to consider possible causes and to summarize information on ways to prevent it. Understanding the possible causes of hyperhydricity through a comprehensive consideration of all symptoms will allow us to offer the most universal and effective way to eliminate this physiological disorder.

## 2. Microscopic and Histochemical Features of Hyperhydric Tissues

In hyperhydric plants, anomalies in the structure and chemical composition of cells and tissues were noted. Changes in the epidermis of leaves of various species cultivated at high humidity *in vitro* were observed in both cuticular and guard cells. There is a significant deformation of the epidermal and glandular trichomes [26]. Hyperhydric leaves do not have a distinct palisade parenchyma; their stomata and parenchymal cells are distorted and have extensive intercellular spaces. *In vitro*, plant stomata lack a closure mechanism in response to darkness, exposure to carbon dioxide, abscisic acid, hypertonic solutions, and water deficiency, which may be the main cause of water loss and death during acclimatization to low relative humidity [5,7,40]. Histochemical studies of guard cells of hyperhydric shoots showed a decrease in pectin level, cellulose level and cutin level, as well as a decrease in the thickness of the cell wall, damage to the integrity of membranes and chloroplasts, a decrease in mitochondrial number, and high vacuolization [12,26,41]. Hyperhydric plants are characterized by a reduction in the number of layers of palisade parenchyma and an increase in cell volume. Palisade parenchyma is more similar to spongy parenchyma [42]. Thus, in shallots and apple trees, hyperhydric plants’ cells were larger than normal ones, approximately 13 and 5 times, respectively [43,44]. Reduction of the palisade parenchyma in hyperhydric leaves was noted in such species as *Helianthus annuus* [45], *Cynara scolymus* [46], *Dianthus cariophyllus*, *Castanea sativa* [4]. The leaves of hyperhydric plants also contain fewer chloroplasts per cell, and chloroplasts are characterized by changes in the thylakoid structure [44]. Cell ultrastructure of *Beta vulgaris* hyperhydric shoots showed denatured chloroplasts in a myxoplasm mass formed by the damage to the tonoplast and the mixing of the cytoplasm with the vacuolar juice. The nuclei were picnotic, presenting paranucleolar corpuscles [21].

Hyperhydric stems are smaller in diameter and show atrophy of the cortical and cork layers, as well as hypolignification of the vascular system. The stem cells of hyperhydric shoots are characterized by thin walls, large intercellular air spaces, a reduction of conductive tissue, a lack of sclerenchymal tissue, and hydration of the core and cortical layer [45]. In hyperhydric carnation shoots, the absence of a procambial strand and normal organization of vascular bundles were noted [4]. Biochemical features of tissues in hyperhydric plants are associated with anatomical and morphological disorders. Hyperhydric plants are characterized by high humidity associated with the accumulation of water in the intercellular spaces [9,47]. In hyperhydric *Arabidopsis* plants, the volume of apoplastic air decreased from 85% to 15%. It was related to water saturation in the apoplast and led to a reduction in gas exchange [48]. Another symptom of hyperhydricity is a decrease in lignin content, which may be due to the low activity of enzymes involved in the biosynthesis of its precursors (flavones and chalcones) and their polymerization [43]. There was also a significant decrease in the activity of enzymes involved in phenolic metabolism, such as phenylalanine ammonium lyase (PAL), cinnamate 4-hydroxylase (C4H), and ionic peroxidase, resulting in a reduced content of phenolic compounds in hyperhydric plants [2,43,49]. The roles of lignin and the phenylpropanoid pathway in hyperhydricity development were confirmed. Exogenously applied p-coumaric acid, a precursor for lignin biosynthesis, reduced symptoms of hyperhydricity, decreased the percentage of the apoplastic water, and increased the total lignin content in wild-type and mutant plantlets of *Arabidopsis thaliana* with less lignin content (the mutation in the gene encoding C4H) [49].

BA positively affects lignification, increasing the activity of peroxidases. For *Brachypodium distachyon* shoots, it was shown that BA causes hyperhydricity but does not alter the lignin/cellulose ratio, which is typical for hyperhydric plants. This data has demonstrated that the gas exchange disrupted the lignin pathway to a greater extent compared to the exogenously applied cytokinin [50]. An increase in glutamate dehydrogenase activity leads to a slowdown in the biosynthesis of amino acids [2]. Together, these processes may be responsible for the low protein content in hyperhydric plants.

## 3. Causes and Physiological Bases of Hyperhydricity

Some changes in the metabolic processes of hyperhydric plants are similar to stress effects. This reaction can be considered an adaptive response to several stress factors simultaneously, including wounding of the explant and osmotic shock due to the penetration of the medium into the intercellular space. Too much liquid or soft medium in excess and a high concentration of cytokinins can also lead to hyperhydricity. High relative humidity and gas accumulation (mainly CO_2_ and ethylene) in the atmosphere of confined culture vessels are also factors in hyperhydricity development [2]. Thus, it is possible to single out the factors associated with hyperhydricity *in vitro*: (1) the composition of the medium, the presence of growth regulators, and the type of gelling agent used [18,19,38,51]; (2) cultivation conditions, the presence of aeration [23,38,52]; (3) the type of explant, its size, age, and genotype [38,53].

There is no consensus on the unambiguous mechanisms of hyperhydricity, despite the many publications in this area. According to the most widespread opinion, growth under stress conditions leads to reactive oxygen species accumulation (ROS) [7,54,55]. Hyperhydricity leads to the generation of O_2_^−1^, H_2_O_2_, and OH mainly in apoplasts, chloroplasts, mitochondria, and the cytosol [56]. It was also found that H_2_O_2_ is localized in the cell walls and intercellular spaces of hyperhydric plants [7].

Violation of cell structure naturally leads to cellular dysfunctions. Changes in the ultrastructure of the cell wall, cell membrane, vacuoles, vascular tissues, and trichomes can lead to abnormal uptake, transport, and evaporation of water [11]. It is possible that fluid excess in the apoplast is the physiological basis for hyperhydricity [9]. Disruption of gas exchange due to fluid in the intercellular spaces leads to hypoxia. Many abnormalities in hydrated tissues are probably associated with damage due to oxidative stress, which is associated with hypoxia [13,44]. Excessive absorption of water can also promote the formation of ROS within the cell, which act on membrane lipids, nucleic acids, enzymes, and other cellular structures. It has been shown that antioxidant enzyme activity, such as catalase, superoxide dismutase, peroxidase, and glutathione peroxidase is higher in the leaves of hyperhydric plants [9,44,57,58]. There is evidence that the excessive activity of NADPH oxidases, peroxidase, and polyamine oxidase is associated with the formation of endogenous ROS in the membranes of hyperhydric plants, and NADPH oxidases play a decisive role [58].

It has been shown that ROS affect the metabolism of ethylene [59], while ethylene can act as an enhancer of ROS accumulation [60]. To date, based on the available data, it can be assumed that the accumulation of ROS and ethylene are the key factors causing hyperhydricity, but the primary signal is unknown and requires further study. There is an assumption that the primary signals that induce stress and, as a consequence, the accumulation of ethylene and ROS may cause damage to the explant, excessive moisture *in vitro*, the effect of endogenous growth regulators, and an imbalance of mineral components in the medium [11]. In this case, ROS act as a secondary messenger in the induction of the signaling cascade. If ROS are not eliminated by the function of antioxidant systems, their excessive accumulation leads to cell damage, primarily at stomata. Abnormalities in the structure of the stomata lead to disruption of their function and an imbalance in the absorption and loss of water in the tissues. The accumulation of water in tissues ultimately leads to hypoxia and an increase in the production of ROS, which lead to more serious damage to cellular structures, which are manifested as symptoms of hyperhydricity.

According to another opinion, excess ammonium ions are quickly absorbed from the medium, which can increase the intake of carbohydrates in an amount sufficient to reroute them from the metabolic pathway of lignin biosynthesis. It is known that the deficiency of lignin and cellulose in tissues is a consequence of a violation of the nitrogen and carbon ratios after an increase in nitrogen absorption. The lack of these two substances leads to a decrease in the cell wall pressure and, thus, the absorption of water is increased, which can cause hyperhydricity [2].

Although a variety of approaches to eliminating hyperhydricity have been described, its mechanism is still poorly understood at both the molecular and genetic levels. RNA-seq analysis showed that normal and hyperhydric *Prunus persica* leaves differed in the expression of more than 300 genes. The first 30 differently expressed transcripts are associated with photosynthesis, metabolic pathways in response to abiotic stress, and cuticle development. These data are consistent with the fact that this disorder has a significant effect on photosynthesis, oxidative phosphorylation, and the synthesis of pyridine nucleotides [61]. Proteomic analysis of normal and hyperhydric carnation shoots showed quantitative differences in 40 proteins, most of which are involved in photosynthesis (16%), primary metabolism (16%), and RNA processing (18%). A certain amount of these proteins is also involved in secondary metabolism (2–5%) [57]. Thus, individual proteins as well as general processes identified in these studies can be markers for increasing hyperhydricity resistance in *in vitro* cultivated plants.

Differential expression analysis performed on normal and hyperhydric leaves of *Prunus persica* showed that the expression patterns of miRNAs in hyperhydric leaves differed from normal ones. It is assumed that miR5021 and miRnovel2 may play a critical regulatory role in the development of hyperhydricity symptoms in relation to the stress response [62]. The role of plant microRNAs in responses to various types of stress is known. The level of transcription of most of the precursors of these microRNAs in hyperhydric plants is reduced. For example, a decrease in the expression level of miR398 may be associated with the development of resistance to hyperhydricity. It was previously established that a low level of miR398 expression plays an important role in resistance to oxidative stress [63].

To explore the epigenetic mechanism of hyperhydricity development, whole-genome bisulfite sequencing was performed on normal plantlets, hyperhydric plantlets, and hyperhydric plantlets of *Dendrobium officinale* cultured with K_2_SiO_3_. The methylation level in the CHH (where H = A, C, or T nucleotides) context decreased with hyperhydricity, while the exogenous addition of silicon increased the methylation level of hyperhydric plantlets [64]. Whole-genome bisulfite sequencing data has demonstrated that the global DNA methylation level also decreased in hyperhydric seedlings of *Arabidopsis thaliana*. The hyperhydric seedlings displayed CHH context demethylation patterns in the promoters of genes for 1-aminocyclopropane-1-carboxylase synthase (ACS1) and 1-aminocyclopropane-1-carboxylic acid oxidase (ACO1), key enzymes in ethylene biosynthesis, resulting in upregulated expression of both genes and increased ethylene accumulation. Thus, DNA demethylation is a key switch in the activation of genes for ethylene biosynthesis to enable signal transduction, which may subsequently influence aquaporin phosphorylation and stomatal aperture, eventually causing hyperhydricity [65].

## 4. Influence of Medium Composition and Culture Conditions on Hyperhydricity Level

### 4.1. Influence of Nitrogen Ions

The content of macroelements in medium has a significant effect on the morphological and anatomical structure of shoots *in vitro*, as well as on their chemical composition. Early studies show that the content of nitrate and ammonium nitrogen in the medium affects the level of hyperhydricity [66,67]. Ammonium is known to be absorbed faster than other nitrogen sources, such as nitrates. It has been reported that hyperhydricity, nitrate nitrogen levels, and total nitrogen content increase linearly with an increase in ammonium nitrogen in the medium for *Amelanchier arborea* [67]. Thus, a decrease in the concentration of NH_4_NO_3_ in MS medium by 2-3 times reduces the level of hyperhydricity in *Satix babylonica*, *Prunus avium* [66], *Phoenix dactylifera* [34], and *Aloe polyphylla* [18]. In carnations, a decrease in the NH_4_ to NO_3_ ratio improved shoot morphogenesis. It is also said that lignification and hyperhydricity decrease in plum, cactus, and willow when using media with a low nitrogen content [66].

### 4.2. Influence of Other Ions

The use of the medium with a reduced content of mineral salts or MS medium with a two-fold reduced content of all components leads to a decrease in hyperhydricity of clove and cucumber shoots [12]. An increase in the concentration of calcium in the medium was an effective measure for eliminating hyperhydricity in *in vitro* culture for some tree species [68]. However, there are conflicting data, according to which an increased calcium content can disrupt lignification with the formation of peroxidase and induce callose deposition. A decrease in the calcium content in the medium for carnation cultivation led to an increase in the content of chlorophyll and wax in the leaves. It is known that calcium ions are involved in tubulin polymerization and are associated with the activity of β-1,3-glucan synthase in stimulating callose deposition [4]. It is possible that in addition to the effect of Ca on the release of peroxidases from the cell walls, which are associated with hypolignification, the availability of Ca ions causes disorientation of microfibrils and deposition of callose in the mesophyll and epidermis, including guard cells [4]. Thus, hypolignification, disorientation of microfibrils, and the accumulation of callose instead of cellulose affect tissue vitrification. In the tissues of hyperhydric plants, intercellular interaction is reduced. Calcium is the most important cementitious element. However, it has been demonstrated on shallot shoots that hyperhydric tissues contain higher amounts of calcium and free sugars than normal ones. Perhaps this is due to the accumulation of these substances in cells and not in the intercellular space [43]. The results of scanning electron microscopy showed that cells of normal plants are compactly located relative to each other, while in the tissues of hyperhydric plants, extensive intercellular spaces are observed [43]. Low potassium content in the medium is also associated with hyperhydricity [69]. It is assumed that the lack of potassium ions disrupts plant metabolism, leading to the accumulation of sugars and amino acids due to an insufficient amount of energy, which entails a decrease in protein biosynthesis levels.

### 4.3. Influence of Hormones and Growth Regulators

A close relationship between the use of growth regulators and *in vitro* disruption of seedling development has been demonstrated in many studies, although the effects of hormones on abnormal shoot development are still limited and contradictory. Perhaps high concentrations of hormones and growth regulators in the medium stimulate the development of anomalies due to an imbalance in endogenous hormone level. The increased content of internal cytokinins in the tissues of hyperhydric shoots compared to normal ones was confirmed by several authors [69,70]. The dependence of the degree of hyperhydricity on the concentration of cytokinins in the medium was observed in *in vitro* cultivated shoots of aloe, chlorophytum, bluehead, garlic, cloves, apples, pears, and gerbera [15,19,29,71,72,73]. For many crops (carnation, melon, onion, apple, and pine), it has been shown that the presence of BA in the medium at both relatively low (4.4 µM) and high (8.9 µM) concentrations can cause hyperhydricity [74]. According to one of the hypotheses, a high concentration of cytokinins enhances ethylene biosynthesis, which can cause hyperhydricity as well as impaired shoot proliferation and apical necrosis [15,75].

Thus, reducing the concentration of cytokinins in the medium is the optimal strategy for reducing hyperhydricity in *in vitro* culture. Replacement of BA with kinetin, which is considered a less potent cytokinin, also led to a decrease in hyperhydricity symptoms in apple and pear shoots. With long-term cultivation of shoots, it is considered optimal to alternate hormone-free media and media containing cytokinins, since a decrease in their concentration leads to a decrease in the coefficient of clonal micropropagation [74].

Thidiazuron (TDZ)-induced hyperhydricity of shoots was reported for several plant species, such as *Pluchea lanceolata* (>2.3 μM TDZ) [76], *A. polyphylla* (2.5 μM TDZ) [19], and *Arbutus unedo* (18.2 μM TDZ) [77]. Regenerated shoots of *Annona glabra* on medium supplemented with 4.5 μM TDZ demonstrated a significant reduction of chlorophyll a content and the formation of abnormally shaped chloroplasts [78]. TDZ in relatively low concentrations (0.04–0.4 μM) also induced hyperhydricity in *D. caryophyllus*, decreased the photosynthetic membrane stability, and caused the destruction of photosynthetic pigments [79]. TDZ treatment can increase ethylene biosynthesis. It has been proposed that ethylene accumulates as a negative by-product of the TDZ-mediated metabolic cascade [80].

Using meta-topoline (mT), a derivative of BA, can solve the hyperhydricity problem in a number of plant species. Adding mT in place of BA to the medium caused a significant decrease in the hyperhydricity of apple shoots [81], micropropagating *Eucalyptus* [82], *Corylus colurna* shoots [83], *Syzygium cumini* [84], and other herbaceous and woody plant species.

### 4.4. Effect of Gelling Agents

It is believed that the effect of a gelling agent on hyperhydricity in plants depends on its type and brand [68]. Commercially available agar varies in quality and purity. The lack of hyperhydricity has been demonstrated while using several commercially available brands of gelling agents (Difco™ Agar, Fluka^®^ Analytical, Agar-Agar and Phytagel) [85]. Gelling agents based on gellan gum (Gelrite^TM^) contain fewer impurities, which is why they are widely used in plant cell and tissue culture. To obtain a medium of the required density, they are used in smaller quantities than agar. However, according to some studies, the use of gellan gum does not reduce hyperhydricity [69,86]. Apparently, the physical structure of gels based on gellan gum promotes a more intense absorption of substances associated with a high level of hyperhydricity: water, cytokinins, and ammonium ions [86]. There is evidence of an increased content of endogenous cytokinins in plants grown on a medium with gellan gum in comparison with plants cultivated on an agar medium [71]. Increasing the concentration of agar in the medium is an effective measure for eliminating hyperhydricity *in vitro* in many species [6,15,19,87]. But the efficiency of clonal micropropagation can be reduced since the availability of various components of the medium, in particular cytokinins, decreases.

### 4.5. Influence of Culture Vessel Atmosphere

The growth and development of plants are affected not only by the mineral composition of the medium but also by the gas composition of the environment [52]. As a rule, cultivation in a closed vessel is carried out at constant temperature, high humidity, and variable CO_2_ concentration and is accompanied by changes in the composition of the medium and the possible accumulation of toxic metabolic products. In the space of unventilated vessels, such substances as ethylene, CO_2_, acetaldehyde, and ethanol accumulate. The gases circulating in a closed culture vessel can slow down the growth of plant tissues and cause unwanted morphogenetic and physiological changes [4]. Of these, ethylene and CO_2_ have the most significant effect on cell and tissue culture. Some studies have shown that plants with symptoms of hyperhydricity are characterized by increased levels of ethylene [88]. In hyperhydric plants, the expression of the ACO gene, which encodes an enzyme involved in ethylene biosynthesis, is enhanced [9]. Eliminating excess ethylene is an effective anti-hyperhydricity measure. The use of artificial ventilation during clonal micropropagation makes it possible to obtain morphologically and physiologically normal plants without symptoms of hyperhydricity, since it leads to a decrease in the accumulation of CO_2_ and ethylene [6,20,52]. It is also possible to use absorbents and inhibitors of ethylene biosynthesis [26].

### 4.6. Influence of Light

The influence of light on the development of hyperhydricity has not been sufficiently studied. According to Muneer et al. (2016), red and blue LEDs reduced hyperhydricity to control levels in carnation shoots, maintaining thylakoid protein composition and antioxidant defense mechanisms [57]. For *Lippia grata* Schauer shoots, the combination of partial sealing with red/blue LED at a 1:1 light ratio resulted in a 61% reduction of hyperhydric leaves in comparison with traditional conditions [89]. The quality of *Pyrus communis* L. shoots cultivating under far-red LED was inferior because of hyperhydricity and chlorosis, as indicated by their low chlorophyll and carotenoid content [90].

### 4.7. The use of Exogenous Additives

A common approach to reducing the level of hyperhydricity *in vitro* is the use of various exogenous additives. These can be inhibitors of the biosynthesis of phytohormones, moisture and gas absorbers, antioxidants, or precursors of the biosynthesis of biologically active substances.

A decrease in kinetin concentration in combination with the addition of adjuvants such as coconut milk, phloroglucinol, and casein hydrolyzate reduced the incidence of hyperhydric shoots in *Eryngium foetidum* L. [29].

The addition of 50 μL of salicylic acid, 250 μL of ascorbic acid, 10 mM spermidine, and 50 μM of hydrogen peroxide per liter of the medium contributed to a decrease in hyperhydricity in garlic shoots; however, hyperhydricity was aggravated by high concentrations of hydrogen peroxide and spermidine. According to the same study, mannitol had no effect on hyperhydricity, while polypropylene glycol (PEG) induced the formation of hyperhydric shoots [15]. In another study, PEG (0.1%) in combination with NaCl (0.2%) was used as osmotic stress-inducing agents to solve the problem of hyperhydricity during *in vitro* propagation of *Agave sisalana.* Plants subjected to the treatment of these additives have demonstrated the minimum hyperhydricity symptoms [17]. The use of silica gel to absorb excess water in a cultural medium was ineffective and promoted the development of abnormal shoots of *Agastache foeniculum* [16].

It was noted that silver ions can bind to ethylene receptors, thereby preventing the transduction of the ethylene signal [26]. AgNO_3_ and Ag nanoparticles are widely used as additives to the nutrient medium to reduce the level of hyperhydricity, for example, in sunflower [30], thyme [39], melon [24], Rohida tree [91], and other species [22,26]. The gene expression pattern of ACS1 and ACO1 showed reduced expression after Ag treatment of *Dianthus chinensis* [26,28,92] and *Tecomella undulata* [93] microshoots. CoCl_2_ is also an inhibitor of ethylene biosynthesis [26]. The combination effect of AgNO_3_ (10 µM) and CoCl_2_ (5 µM) showed a complete reversal of hyperhydricity in *Dianthus chinensis* shoots [28].

The positive effect of silicon on the recovery of hyperhydric carnation shoots has been demonstrated. The addition of K_2_SiO_3_ to the medium at concentrations of 1.8 mM and 3.6 mM slowed down lipid peroxidation and promoted the restoration of stomatal functions. The results of proteomic analysis indicate the active participation of silicon in various metabolic processes aimed at the restoration of hyperhydric shoots [25,94]. It is known that silicon in the medium in the form of potassium, sodium, and calcium silicates improved shoot growth, promoted an increase in biomass, and an increase in the stability of cells, tissues, and organs in *in vitro* culture [95,96,97]. The role of silicon in physiological responses to various types of stress in plants is also known. It maintains stomatal structure and water balance and regulates the activity of antioxidant enzymes under conditions of drought and salt stress [97,98]. It is noteworthy that silicon accumulates in the apoplast, which is most susceptible to the action of ROS during hyperhydricity.

The addition of exogenous polyamines (spermidine and putrescine) to the medium contributed to an increase in the total content of phenolic compounds, an increase in antioxidant activity, and a decrease in the level of hyperhydricity by 50% in apple shoots [33]. Supplementation of 5 µM spermine in MS medium significantly reduced the hyperhydricity symptoms of *Dianthus chinensis* shoots [27]. Maintaining a high level of endogenous polyamines is necessary for osmoregulation of cells, maintenance of antioxidant activity, and regulation of plant growth [33].

The addition of L-glutamine and adenine sulfate to hormone-free media promoted the development of iris shoots without signs of hyperhydricity during long-term cultivation [86].

## 5. Conclusions

Hyperhydricity is a physiological disorder that is influenced by many related factors. A liquid medium promotes the absorption of excess moisture, which accumulates in the intercellular spaces and disrupts gas exchange, causing hypoxia and oxidative stress. Ammonium ions in high concentrations lead to disruption of plant metabolism, causing anatomical disorders, and an excess of cytokinins enhances ethylene biosynthesis.

Using one or more of the above approaches to eliminate hyperhydricity is not always effective. Different types of plants may require specific media and cultivation conditions. In this regard, the hyperhydricity of individual species should be studied individually and comprehensively, and approaches to its elimination should be complex and species-specific.

## Figures and Tables

**Table 1 plants-11-03313-t001:** Strategies to control hyperhydricity symptoms during cultivation *in vitro*.

Plant Species	Measures to Control Hyperhydricity	Reference
*Agastache foeniculum*	Ventilation of culture vessels (using paper to cover the cultural vessels), increasing the concentration of agar in the medium	[16]
*Agave sisalana*	Adding polypropylene glycol (0.1%) in combination with NaCl (0.2%) as osmotic stress inducing agents	[17]
*Allium sativum*	Adding 50 μM salicylic acid, 250 μM ascorbic acid, 10 μM spermidine, and 50 μM hydrogen peroxide to the medium	[15]
*Aloe polyphylla*	Using agar as a gelling agent, reducing the concentration of cytokinins, ventilation of vessels for cultivation (using modified lids with a hole covered with polyester or cotton mesh)	[18,19,20]
*Beta vulgaris*	Using culture medium prepared with deuterium- depleted water (25 ppm deuterium)	[21]
*Caladium bicolor*	Addition of silver nitrate at 7.5 µM	[22]
*Castanea sativa*	Using agar as a gelling agent, increasing light intensity, ventilating culture vessels	[23]
*Citrullus lanatus*	Adding AgNO_3_ and Ag_2_S_2_O_3_ to the medium	[24]
*Dianthus caryophyllus*	Adding silicon to the medium	[25]
*Dianthus chinensis* L.	Adding AgNO_3_ to the culture medium; adding 10 µM AgNO_3_ in combination with 5 µM CoCl_2_ to the culture medium; supplementation of 5 µM spermine in MS (Murashige and Skoog) medium	[26,27,28]
*Eryngium foetidum* L.	Reducing the concentration of kinetin, adding adjuvants to the medium, such as coconut milk, phloroglucinol and casein hydrolysate	[29]
*Helianthus annuus* L.	Adding AgNO_3_ to the medium	[30]
*Lippia grata*	Ventilation of culture vessels (partial sealing treatments)	[31]
*Lycium ruthenicum*	Treatment of starvation and drying combined with 30 μM AgNO3	[32]
*Malus domestica*	Using gellan gum as a gelling agent, adding 10–5 M spermidine, arginine or ornithine to the culture medium	[33]
*Phoenix dactylifera*	Using modification at a ratio of NH^4+/^NO^3-^ at 10:15 (825:1425 mg/L) of MS medium	[34]
*Pistacia vera*	Using methoxy topoline-riboside (MemTR) and meta-topoline-riboside (mTR) as an alternative to benzyladenine (BA)	[35]
*Pyrus communis*	Using MemTR and mTR as an alternative to BA	[36]
*Salvia santolinifolia*	MS medium modification with NH_4_NO_3_ (412 mg/L), KNO_3_ (475 mg/L) and CaCl_2_·2H_2_O (880 mg/L)	[37]
*Scrophularia yoshimurae*	Using gellan gum as a gelling agent, ventilation of vessels for cultivation (using dispense paper)	[38]
*Thymus daenensis*	Adding salicylic acid and colloidal silver nanoparticles to the medium	[14,39]

## Data Availability

Not applicable.

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
