# Peer review of "Hyperhydricity in Plant Tissue Culture"

_plants, 2022, doi:10.3390/plants11233313_

Round 1

Reviewer 1 Report

The manuscript “Hyperhydricity in plant tissue culture” is a useful review for biotechnology researchers. The manuscript describes hyperhydricity symptoms in vitro plants and possible solutions. It could be a relevant and interesting research, however, should be returned to the authors for a thorough rewrite.

I found the manuscript difficult to follow because of incorrect use of English. I started highlighting on the pdf file those words/sentences I found incorrect, but I did not do it through the whole of the manuscript.

In addition, the references are not very update. I have given below some more recent interesting references that should be included in the manuscript. These are just some examples of what is present in the literature. Therefore, I suggest Authors to update the review and the table after extensive bibliographical research.

 Bayraktar, M., Hayta-Smedley, S., Unal, S., Varol, N., & Gurel, A. (2020). Micropropagation and prevention of hyperhydricity in olive (Olea europaea L.) cultivar ‘Gemlik’. South African Journal of Botany128, 264-273.

Gantait, S., & Mahanta, M. (2022). Hyperhydricity-induced changes among in vitro regenerants of gerbera. South African Journal of Botany149, 496-501.

Jan, T., Gul, S., Khan, A., Pervez, S., Noor, A., Amin, H., ... & Ullah, H. (2021). Range of factors in the reduction of hyperhydricity associated with in vitro shoots of Salvia santolinifolia Bioss. Brazilian Journal of Biology83.

Lotfi, M., Bayoudh, C., Werbrouck, S., & Mars, M. (2020). Effects of meta–topolin derivatives and temporary immersion on hyperhydricity and in vitro shoot proliferation in Pyrus communis. Plant Cell, Tissue and Organ Culture (PCTOC)143(3), 499-505.

Nikam, T. D., Mulye, K. V., Chambhare, M. R., Nikule, H. A., & Ahire, M. L. (2019). Reduction in hyperhydricity and improvement in in vitro propagation of commercial hard fibre and medicinal glycoside yielding Agave sisalana Perr. ex Engelm by NaCl and polyethylene glycol. Plant Cell, Tissue and Organ Culture (PCTOC)138(1), 67-78.

Sreelekshmi, R., Siril, E. A., Muthukrishnan, S. (2022). Role of biogenic silver nanoparticles on hyperhydricity reversion in Dianthus chinensis L. an in vitro model culture. Journal of Plant Growth Regulation41(1), 23-39.

The table 1 needs to be improved to make it easier to understand; changes have been made directly to the pdf text and the reference numbering is completely wrong. If the table is correctly placed, the reference numbering must be consequential to that of the text.

Reviewer 2 Report

The paper needs minor revision

Author Response

Thank you very much for positive evaluation of our work.

Round 2

Reviewer 1 Report

The second version of the manuscript "Hyperhydricity in plant tissue culture" has been improved, especially regarding the use of English. However, I still recommend that the manuscript is revised by a native person or professional editor.

The bibliography has been updated, correctly inserted in the text. 

I have no corrections to add to the manuscript.